# Self-Injurious Behavior in Community Youth

**DOI:** 10.3390/ijerph18041955

**Published:** 2021-02-17

**Authors:** Yeonkyeong Son, Sojung Kim, Jong-Sun Lee

**Affiliations:** 1Department of Psychiatry, Hanyang University Medical Center, Seoul 04763, Korea; ooee2728@naver.com (Y.S.); sojungclair@gmail.com (S.K.); 2Department of Psychology, Kangwon National University, Chuncheon-si 24341, Korea

**Keywords:** self-injurious behavior, community youth, risk factor

## Abstract

The rapid increase of self-injurious behavior among Korean adolescents, and its reckless spread on social media, has driven the necessity to study this behavior. The present study investigates the characteristics of self-injurious behavior among adolescents in local communities, and the psychological factors affecting such behavior. Questionnaires were administered to 516 sixth graders in elementary school and first to third graders in middle school of both genders, nationwide. They measured the prevalence and characteristics of self-injurious behavior and the relevant psychological factors, such as levels of depression, anxiety, and self-esteem. Furthermore, group differences were assessed for self-injury experience and the characteristics relevant to self-injurious behavior. In addition, this study performs logistic regression to explore the risk factors predicting self-injurious behavior. In all, 166 participants (32.2%) reported self-injury, with a higher rate of self-injury in female students than in male students. Although the study finds high rates of mild forms of self-injury, such as “biting”, “pulling hair,” and “hitting self”, it also finds relatively high reports of more risky methods, such as “cutting or carving”. The logistic regression shows a significant effect of the negative self-image sub-factor of depression (CDI) and oversensitivity and physical and sleep problems sub-factors of anxiety (RCMAS) on self-injurious behavior. The rates of self-injury were higher in female participants than in male ones, and adolescents in local communities reported higher rates of mild forms of self-injury than the moderate/severe forms. The results of this study suggest that early screenings and interventions should be conducted through evaluation of self-image and emotional stability of early adolescents to hinder the risk of self-harm.

## 1. Introduction

Self-injurious behavior is a broad class of behaviors that directly and deliberately harm one’s body. These behaviors include non-suicidal self-injury (NSSI), in which one directly and deliberately harms one’s own body tissue without the intent to die [1]. NSSI includes cutting or carving one’s skin with a sharp object, hitting oneself, and picking one’s wounds. In addition, banging, burning, and substance abuse are also common forms of NSSI [2]. Previous research has categorized the severity of NSSI as mild, moderate, and severe according to their forms. Mild forms of NSSI include hitting oneself, picking one’s wounds, and banging, while moderate/severe forms of NSSI consist of cutting, carving one’s skin, burning, and substance abuse [3]. Self-injury tends to begin in early adolescence, typically in ages 12–14 [4]. Meta-analysis studies have reported a 2.9–69.6% lifetime prevalence of NSSI in children and adolescents, varying with samples and methods of evaluation. They estimate the overall lifetime prevalence at 22.1% [5]. Self-injurious behavior is highly correlated with various psychiatric symptoms. For example, symptoms of depression and anxiety disorders were more associated with self-harm, especially those who did self-harm had a higher number of borderline personality disorder (BPD) symptoms than those who did not [6]. Therefore, such behavior requires care and attention, as repetitive self-injury desensitizes an individual to physical pain and it can increase the risk of suicide. Large-scale cohort studies conducted in the United States and United Kingdom found that adults who deliberately injured themselves had a suicide rate 30 times higher than that of the general population [7,8].

Recent studies conducted in Korea investigating self-injurious behavior in middle and high school students report varying degrees of lifetime prevalence, ranging from 8.8% to 22.8%; some studies find that the age of onset of self-injury is 12.4~13.8, which is similar to studies conducted overseas [9,10,11,12,13]. As a result of the Adolescent Personality and Mental Health Problems Screening Questionnaire conducted by the Korean Ministry of Education for all middle and high school students in Korea in 2018, the number of first-year middle school students who reported that they had engaged in self-harm was about 40,500/514,710 (7.9%), and about 30,000/514,710 (6.4%) in the first year of high school reported self-harm experiences [14]. An analysis of the number of counseling cases at 230 youth counseling and welfare centers nationwide in Korea identified 27,976 cases of counseling for adolescents who engaged in self-injurious behavior in 2018, which was more than three times higher than that of the previous year (8352). During the same period, the number of suicide-related counseling also doubled from 23,915 (2017) to 43,238 (2018) [15]. In addition, data from the Adolescent Personality and Mental Health Problems Screening Questionnaire for the last four years revealed that the number of students in the suicide risk group in 2018 was 23,324, which is an increase of 270% compared to 2015 [16]. Considering that the survey was conducted at the school level, the proportion of students who have engaged in self-harm may have been less reported. Nevertheless, the fact that the proportion of students at risk of self-harm and suicide has increased sharply in the last few years, reflecting the recent issue of youth self-harm as a serious problem in Korea.

The degree of self-injurious behavior can be classified as mild or moderate based on the severity and frequency of self-injury [3]. The characteristics vary, and depend on the number of methods used or the age of onset. In general, the severity of self-injury tends to increase, as both the number of self-injuries and methods used increase [1,17], and with the decreasing age of onset [18]. Previous research in Korea reports that mild forms of self-injurious behaviors, such as “biting,” “hitting,” “pulling hair,” and “scratching skin,” are most common in adolescents in local communities, and 13.7–16.8% of the participants report moderate forms of self-injury, such as “cutting with a sharp object” [9,12,13]. Lee [11] reports a moderate degree of self-injurious behavior in 67.0% of middle-schoolers. Prior research shows that self-injury in adolescents helps them regulate burdensome negative emotions, as a coping strategy to deal with emotional difficulties [19,20]. Early intervention is essential, because repetitive self-injurious behavior in the early stages of development can become the main coping strategy of adolescents when they face unpleasant emotions or situations. With adolescent self-injury in local communities becoming a serious problem in society, an increasing number of studies in Korea are now focusing on such behavior [21]. However, there remains a lack of foundational research investigating the conditions and characteristics of Korean adolescent self-injurious behavior.

Recently, adolescents in Korea have begun taking pictures of self-injurious behavior and uploading them on social media or forming cyber communities of self-injury, such as “ja-hae-gye” (self-injury group) or “ja-hae-ruh” (individual with self-injurious behavior) [21]. Self-injury occurs to regulate emotions or fulfill social purposes, such as manipulating others or avoiding responsibilities; however, peer pressure is a strong factor that influences adolescents to choose self-injurious behavior as a coping strategy over others [22]. In a longitudinal study targeting adolescents, the number of self-injuries of the closest friend significantly predicted the number of self-injuries in early female adolescents 11 months later [22]. Another study that examined clinical samples found that 82.1% of adolescents who harm themselves have friends who also self-harm [23]. The socialization effect can explain deviant behaviors in adolescents, such as alcohol and drug use, sexual behavior, and other acts of delinquency, including health-risk behaviors [24]. The socialization effect states that certain behaviors or attitudes of peers strongly affect the behaviors and attitudes of other peers within the same group (for a review, see [25]). Peer relationships provide context for the socialization of behaviors, and in recent years, such effects have been generalized to include socialization of some extreme forms of behavior, such as self-injury [22,26]. Considering the trend of self-injurious behavior of adolescents in real life and on the internet, self-injury is now a sub-culture of teenagers that helps them form a sense of fellowship in addition to its traditional use as a maladaptive coping strategy. Self-injurious behavior for early adolescents in the current society seems to have a different function than before. Therefore, it is necessary to examine these aspects further to understand these emerging characteristics.

Self-injurious behavior generally means deliberately and directly harming one’s body; however, it has two subtypes: suicidal self-injury and non-suicidal self-injury (NSSI), based on whether or not an individual intends to die [27]. Unlike suicidal behavior aimed to kill oneself, NSSI accompanies multiple purposes other than death. The affect regulation model is the most empirically supported among the theories explaining the functions of NSSI. It describes that self-injurious behavior occurs to regulate emotions that one finds very hard to handle [28]. According to the experiential avoidance model (EAM) proposed by Chapman et al. [19], self-injurious behavior is maintained by negative reinforcement, as it reduces or terminates unwanted negative emotions. A strong urge to avoid elevated emotional arousal can occur, especially when one tends to feel overwhelmed by strong emotional reactivity or experiencing emotional arousals, as unpleasant stimuli, due to low distress tolerance. When individuals lack effective emotion regulation strategies or fail to use one due to high arousal levels when experiencing unwanted emotion, they might choose a maladaptive regulation strategy such as self-injuring to reduce unpleasant emotional arousals quickly. If self-injurious behavior is negatively reinforced repeatedly, it can be habituated as a response to the stimuli that cause negative affect, and eventually, it can function as an automated and conditioned response to emotional arousals.

Previous research shows that individuals who self-injure generally experience higher levels of negative affect than those who do not injure themselves [29,30]. Similar to adults, adolescents perform self-injurious behavior primarily to soothe tension and emotional pain and avoid negative affect [31,32]. One study that examined self-injuring adolescents in local communities reported that 73.0% of the sample self-harms to obtain relief from a terrible state of mind [33]. Such results support the thesis that affect regulation is the key component of self-injurious behavior, as EAM also asserts.

In addition, the integrated theoretical model of NSSI proposed by Nock [34], which is based on previous studies, claims that early developmental factors, such as childhood abuse or genetic factors of high emotional and cognitive reactivity, induce internal and external vulnerability within a person, as they function as the distal risk factors of self-injury. Individuals with such vulnerability use maladaptive coping strategies, such as self-injuring, to regulate strong negative emotions induced by stressful events. According to the above-mentioned model, the negative emotional states that a person experiences, function as the ultimate proximal risk factors for self-injury. According to the above-mentioned model, the negative emotional states a person experiences function as the ultimate proximal risk factors for self-injury. However, this model did not depict specific negative emotions as risk factors of self-harm. Previous studies have found a strong relationship between self-injury among adolescents and negative emotions, namely depression and anxiety. We aimed to investigate specific aspects of such negative emotions in association with self-injurious behavior.

The results of preceding research that insist affect regulation is the key function of self-injury and negative emotional states are the proximal risk factors of self-injurious behavior, indicate that further investigations should focus on negative emotion as a critical factor triggering self-injurious behavior. Many studies have identified depression as the risk factor predicting self-injurious behavior in Korea and abroad [10,11,35,36,37,38,39,40,41,42,43,44,45,46,47,48], and anxiety [10,39,40]. They are also strong proximal risk factors for self-injury [43]. Therefore, they are widely accepted as primary negative emotions inducing self-injurious behavior.

However, a foreign study on psychiatric outpatients found no significant difference in the levels of depression between adolescents who injured themselves and those who did not [42]. Another study conducted in Korea reported a similar finding, in which the levels of depression did not differ between the self-injury group and the non-self-injury group [40]. In addition, studies are inconsistent in reporting the relationship between high levels of anxiety and self-injurious behavior [28,38]. Although that part of the research was on clinical samples, such results propose the possibility of difficulties in predicting self-injurious behavior from the total levels of depression and anxiety. Depression and anxiety comprise multi-dimensional factors, including cognitive, emotional, physiological, and behavioral aspects [43,44,45,46,47,48,49,50]. Therefore, further investigation into the sub-factors and distinct predictive powers of self-injurious behavior is necessary to confirm the aspects of depression and anxiety that specifically predict self-injurious behavior.

Moreover, studies show that low self-esteem has a close association with self-injurious behavior in adolescence [51,52]. As self-aversion and the need to punish oneself trigger some self-injuries [29], such behavior can be understood as an attempt to avoid negative emotions induced by unpleasant conditions relevant to negative self-belief or self-awareness [19]. Similarly, low self-esteem, along with negative emotions such as depression and anxiety, can be a risk factor causing self-injurious behavior in early adolescence.

This study aims to investigate self-injurious behavior among adolescents in Korea and the risk factors in such behavior. Specifically, we attempt to understand the characteristics of self-injurious behavior trending among Korean adolescents along with the relevant risk factors to identify the implications for effective prevention and intervention strategies for early adolescents who are at a high risk of such behavior.

## 2. Methods

### 2.1. Participants

The study participants were 516 students residing in Seoul, Gyeonggi-do Province, Gangwon-do Province, and Jeollanam-do Province. The participants included the sixth graders from elementary school and first to third graders from middle school (i.e., seventh to ninth graders) of both genders. The second author’s university approved this study, with IRB (no. HL19C0035). For the schools that approved participation in this study, the researchers provided both students and legal guardians with explanations on the description of this study and its procedures, an assurance of confidentiality of personal information, and a right to participate or withdraw from the study voluntarily. Exclusively to the participants who understood the purpose of this study and presented written informed consent signed by both a student and a legal guardian, a link to the online survey composed of multiple questionnaires was sent via cellular phones. Among the 582 initial volunteers of the study, 527 turned in their answers. After excluding questionnaires with insufficient responses (e.g., questionnaires with several missing items), data from a total of 516 students were used for the final analysis.

Of the 516 students, 212 were male (41.1%) and 261 were female (50.6%); 43 students did not identify their gender. The mean age was 13.74 (*SD* = 0.89), ranging from 11 to 16.

### 2.2. Measures

#### 2.2.1. Functional Assessment of Self-Mutilation (FASM)

This study used the Functional Assessment of Self-Mutilation developed by Lloyd, Kelly, and Hope [53] and translated and validated in Korean by Kwon [54] to measure NSSI. This self-reported scale comprises three parts. The first part comprises 12 items measuring various methods and frequency of NSSI, and whether or not medical treatment was received; the frequency of use of pertinent method in the past year or earlier is rated on a 7-point scale (from 0 time or never to 6 or more times). The second part assesses multiple factors relevant to self-injurious behavior (whether it occurred within the past year, whether there was intent to die, duration of contemplation before self-injury, whether they had consumed drugs or alcohol at the time, the amount of physical pain they felt during self-injury, and the age of onset). The third part consisting of 23 items measures purpose and reasons for participants to engage in NSSI. For analysis, this study only uses the items assessing methods and frequency of self-injury and involving factors pertaining to self-injurious behavior. Cronbach’s alpha for 11 items measuring methods and frequency for self-injury was 0.59.

This study used a total score of frequency of self-injury to analyze correlations with other scales. The severity of different methods of self-injurious behavior was categorized based on previous studies [3,27] and items 1, 4, 6, 11 were classified as moderate, whereas the rest were classified as mild.

#### 2.2.2. Children’s Depression Inventory (CDI)

To assess the severity of depression in children and adolescents, this study used the Children’s Depression Inventory (CDI) developed by Kovacs [55], which Cho and Lee [56] translated and validated in Korean. It is a 27-item questionnaire that uses a 3-point scale (0, 1, 2), and its score ranges from 0 to 54, with higher scores indicating more depressive symptoms. Factor analysis in Kim et al.’s [46] study finds the three-factor structure the most adequate for Korean children; the three factors include negative self-image, interpersonal problems, and negative affect/somatic concerns. Internal consistency was 0.88 in Cho and Lee [56] and 0.85 in Kim et al. [46]. In the current study, internal consistency was 0.84.

#### 2.2.3. Revised Children’s Manifest Anxiety Scale (RCMAS)

This study used the Revised Children’s Manifest Anxiety Scale, developed by Reynolds and Richmond [57] and translated in Korean by Choi and Cho [58]. It is a self-reported scale designed for youths, from elementary school students to high school students. The scale consists of 9 lie items and 28 anxiety-related items, each item being dichotomous to answer as “yes” or “no.” Higher total scores indicate a higher level of anxiety. Park et al. [49] conducted factor analysis on Korean adolescents and revealed that the four-factor model had the best fit; the four factors are worry, oversensitivity, physical/sleep problems, and negative affect/attention problems. Internal consistency in Choi and Cho [50] was 0.81 and in Park et al. [49] was 0.85. This current study has an internal consistency of 0.89.

#### 2.2.4. Rosenberg Self-Esteem Scale (RSES)

To measure self-esteem, this study used the Rosenberg Self-Esteem Scale [59] translated in Korean by Jeon [60] and validated by Lee [59]. This scale, which has 10 items, uses a 4-point Likert scale (1: almost never~4: always), and higher scores reflect higher self-esteem. Internal consistency was 0.79 in Lee [61] and 0.84 in this current study.

### 2.3. Statistical Analysis

After analyzing the descriptive statistics and correlations among key variables, this study analyzed variables accounting for differences in psychological characteristics between the self-injury and non-self-injury group. It also explored the characteristics of self-injurious behavior pertaining to those who having engaged in self-injurious behavior, such as frequency and methods of self-injury. To uncover the risk factors predicting self-injurious behavior, this study conducted binomial logistic regression. All data were processed using SPSS 22.0 [62] at the 0.05 statistical significance level.

## 3. Results

### 3.1. Demographic Statistics

Table 1 shows the demographic statistics of the participants. Among the students who had completed questionnaires, this study categorized those having engaged in self-injurious behavior at least once on FASM as the self-injury (SI) group. Of the 516 participants, 166 were adolescents who having engaged in self-injurious behavior (32.2%) and 350 were adolescent with no experience of self-injury (67.8%). A significant gender difference was found in the self-harm experiences; more female students had self-injured than had male students. More specifically, among the SI group, there were proportionally more females than in the non-SI group.

### 3.2. Characteristics of Self-Injurious Behavior

Further analysis of the data of the SI group (*n* = 166) revealed that they had used an average of 2.04 (*SD* = 1.31) methods, and on average, self-injured 6.22 times (*SD* = 6.50) all their lives until survey completion. In terms of severity, the mean number of mild self-injury was 5.07 (*SD* = 5.35) and of moderate self-injury was 1.15 (*SD* = 2.27). On average, the number of treatments received following the self-injurious behavior was 0.53 (*SD* = 1.65). Table 2 lists the characteristics of methods and the frequency of self-injury. The self-injury group reported higher rates of mild forms of self-injury in the following order: ‘biting,’ ‘hitting’, and ‘pulling hair’.

### 3.3. Descriptive Statistics and Correlational Analysis of Variables Relevant to Self-Injury

To explore the characteristics relevant to self-injurious behavior, this study examined the duration of contemplation before self-injurious behavior, whether alcohol or drugs were consumed, physical pain during self-injurious behavior, whether there was intent to die, and the age of onset. According to the results, in the SI group, most of the participants (72.9%) had not contemplated self-injurious behavior before, and only two participants (1.2%) had used alcohol or drugs during self-injurious behavior. In addition, the majority of participants (90.4%) reported no or mild pain during self-injurious behavior, and 18 participants (10.8%) said they had had a suicidal intention, and the average age of onset of self-injurious behavior was 12.43 (*SD* = 1.57). Details are shown in Table 3.

The study then investigated the relationships among variables pertinent to self-injury. Table 4 shows the results of the correlational analysis. Higher frequency correlated with more methods of self-injurious behavior (*r* = 0.84, *p* < 0.01) and more physical pain (*r* = 0.46, *p* < 0.01). Longer time of contemplation correlated with more physical pain (*r* = 0.44, *p* < 0.01). Moreover, the frequencies of mild and moderate self-injurious behavior were positively correlated with each other (*r* = 0.34, *p* < 0.01); the number of medical treatments showed a positive correlation with only the moderate form of severity (*r* = 0.18, *p* < 0.05) among all levels. The age of onset was positively correlated with the duration of contemplation before self-injury alone (*r* = 0.35, *p* < 0.01).

### 3.4. Descriptive Statistics and Correlational Analysis of Key Variables

Table 5 shows the correlations among all of the key variables included in the analysis explored after dividing the data into the SI group and the non-SI group. All the sub-factors of depression (CDI) and anxiety (RCMAS) revealed positive correlations among them but negative correlations with self-esteem (RSES) in both groups. In addition, in the SI group, the number of self-injurious behavior was positively correlated with every sub-factor of CDI and RCMAS, but was negatively correlated with self-esteem.

Table 6 presents the results of the t-test to verify if there are significant differences in the levels of depression, anxiety, and self-esteem between the two groups. The differences were significant between the two groups in all of the sub-factors of depression and anxiety and self-esteem. Thus, students who had self-injured more than once reported higher depression and anxiety levels for each sub-factor and lower self-esteem.

### 3.5. Risk Factors Relating to Self-Injurious Behavior

To uncover the risk factors affecting self-injurious behavior, this study undertook logistic regression with the sub-factors of depression (CDI) and anxiety (RCMAS) and self-esteem (RSES). The results showed significant differences in the history of self-injury; Table 7 lists these findings. The regression model was statistically significant (*χ^2^* = 87.21, *p* < 0.001), where negative self-image in depression (B = 0.09, *p* < 0.05), oversensitivity (B = 0.18, *p* < 0.05), and physical/sleep problems (B = 0.20, *p* < 0.05) in anxiety were the significant factors affecting self-injurious behavior.

## 4. Discussion

This study investigates the conditions and aspects relevant to adolescent self-injurious behavior in Korea local communities, and confirms specific risk factors predicting self-injury in adolescence. The main results and discussion are as follows.

Of the 516 adolescents that participated in this study, 166 (32.2%) had self-injured, which is higher than the rates of adolescent self-injury in local communities reported in other studies in both Korea and overseas, at approximately 20% [5,9,13]. The rise in self-injurious behavior reported could be because of perception, as something more widely accepted within the peer group today than it was in the past, as such behavior today is not only to reduce negative emotions but also to earn social support and a sense of belonging [63]. A recent study by Kim, Kim, Park, and Yoon [64] revealed that 23.7% of self-injuring elementary schoolers had “deliberately drawn wounds resembling scars of self-injury.” Such a finding reflects the possibility that adolescent self-injury is now recognized as a sub-culture of teenage groups. Thus, early detection and intervention of self-injurious behavior in young adolescents is more crucial than ever.

Further investigation showed that more female students had attempted self-injury than had male students, consistent with previous research in and outside the country [9,38]. The average age of onset was 12.43, which also aligns with preceding research, reporting self-injury first occurring around the ages of 12–14 [4], and with other studies in Korea reporting 12.3–13.8, as the average age of onset of self-injury [9,10,11,12,13]. In addition, this study found that the percentage of mild forms of self-injury, such as “biting,” “hitting,” and “pulling hair,” was higher than other forms. This is similar to the results of previous research conducted in Korea, which claimed that mild forms of self-injury are more prevalent than severe forms among local community adolescents [9,13]. In addition, results of a study on Korean university students found that the proportion of “biting” was 62.1% among self-harm methods, accounting for the highest share among self-harm methods [65]. Moreover, according to a study conducted in the United States, early adult college students tend to use mild forms of self-injury such as “biting,” “hitting,” and “picking” more than the moderate forms [27,66]. Among the students who participated in this study, those who acknowledged self-injury, reported that they self-injured 6.2 times, and used 2.04 methods, on average, over their lifetime. Such results are similar to the findings of Park et al. [13], who reported 6.54 times of self-injury among male high school students, and Ahn and Song [9], which revealed 7.13 times of self-injury among middle and high school students of both genders. Among the previous studies targeting Korean adults, it is difficult to note one that specifically investigated the number of self-injuries; however, if students who begin to self-injure in early adolescence continue to do so until adulthood, the number of self-injuries is likely to be cumulative. Moreover, the rising trend of self-injury on social media among teenagers suggests the formation of a new contextual factor. As sharing self-injurious behavior on social media can possibly normalize self-injury, and continuously encourage such behavior [67], the corresponding group of adolescents may later form a habit and use self-injurious behavior for various purposes, as they enter late adolescence and adulthood. Therefore, it is necessary to continue observing and researching such phenomenon, as recent self-injurious behavior in teenagers may show different aspects than self-injury reported in past studies targeting adults.

This study found that higher number of self-injurious behavior was associated with more methods used and greater physical pain. However, some previous studies reported different findings. An earlier study showed that physical pain tends to become desensitized with repetition of self-injurious behavior [34], and a study on university students also revealed a negative correlation between the number of self-injuries and pain during self-injury [65]. Yet, Park et al. [13], who examined male high school students, reported a finding similar to this current study, stating that as the number of self-injuries increase, physical pain also increases. Such complicated results suggest that young adolescents in local communities who repeatedly injure themselves mildly might have distinct aspects from adults or the clinical group who severely injure themselves. Thus, further investigation is necessary to understand these new features of self-injury in non-clinical adolescents.

In addition, this study found that more than 70% of students who engaged in self-harm did not contemplate this behavior beforehand. This is considered to be consistent with the results of previous studies showing that adolescents in local communities perform self-harm in a more impulsive and habitual manner by various internal and external triggering stimuli [9,13]. Moreover, only two participants in this sample (1.2%) had used alcohol or substances during self-injurious behavior, which can be partly explained by Korea having less access to drugs compared to Western countries. In this study, 10.8% of the SI group reported suicidal intention during self-harm. This is similar to a previous study that reported 7.0% of male and female adolescents in the self-injury group had suicidal intentions [3]. According to a previous study conducted in Korea, notably, 32.0% of female adolescents who engaged in self-injurious behavior reported suicidal intentions [12], whereas 9.1% of the SI group consisting of male adolescents had suicidal intentions [13]. In future studies, it will be necessary to confirm if a significant difference in suicidal intention during self-injurious behavior exists according to age and gender. On the other hand, in this study, no significant cultural differences were found between Western and Asian countries in the form of self-injurious behavior and related characteristics. This may be partly due to the fact that self-injurious behavior was measured in this study using a scale (FASM) developed in a Western country. Therefore, in future studies it will be important to develop scales that are more suitable for Korean culture so that culture-specific self-injurious behavior can be examined.

This study also explored risk factors, showing that the negative self-image sub-factor of depression and oversensitivity and physical/sleep problems sub-factors of anxiety best predicted self-injurious behavior. Few studies have specifically tested relationships between the sub-factors of depression and anxiety and the occurrence of self-injury. However, prior studies frequently report self-punishment as a motivational factor [29,31], which partially explains why individuals harm their physical bodies, as a coping strategy to reduce negative affect. Those who think they are worthless and inferior tend to tolerate more pain during self-injury [68]. Having a negative core belief about the self can reduce the tendency to value or protect one’s physical body and eventually exacerbate psychological resistance to self-injury or suicidal thoughts [34]. Negative perceptions of the self, such as negative self-image, can be a significant factor triggering self-injurious behavior. Furthermore, teenagers in early adolescence are in a crucial period of their lives, where they develop and integrate concepts relevant to self, and thus, the prevention and intervention of self-injury should focus on perceiving the self positively, and developing healthy self-concepts.

The oversensitivity sub-factor of anxiety can be defined as an individual being too sensitive or nervous, become irritated or angered easily with a minor stimulus, or easily hurt from a small event [69]. Research shows that when one shows high physical arousals following emotionally evocative events, one can perceive corresponding stimulus more aversively, and excited arousals can cause difficulties in regulating negative affect [19]. Therefore, the oversensitivity sub-factor of anxiety significantly predicting the occurrence of self-injury can relate to using maladaptive coping strategies, such as self-injury, effortlessly, to terminate an aversive emotional state quickly, when an individual who becomes agitated easily by negative affect experiences a strong unpleasant feeling.

In addition, the failure to recognize and process negative affect can cause physical symptoms. Physical discomfort is considered to be a factor of alexithymia, and it is understood as a major element provoking inappropriate coping mechanisms, such as self-injury, when facing negative emotions [4]. Previous research shows that even after controlling for depression symptoms, sleep problems in ages 12–14 significantly predicted suicide attempts and self-injurious behavior in ages 15–17 [70], and 77.0% of female adolescents who had reported little or no sleep showed repetitive self-injurious behavior after one year [71]. Such results emphasize the importance of considering physical symptoms and sleep problems in preventing self-injurious behavior. Especially, sleep problems not only reflect severe emotional agitation or psychological pain but also can cause unstable affect [72] and can also diminish the ability to use affect regulation strategies [73]. Therefore, sleep problems seem to be closely associated with incidents of self-injury.

In summary, teenagers in early adolescence injure themselves to ease the feeling of inadequacy caused by negative self-perception or to relieve negative emotions induced by multiple psychosocial stresses. These characteristics, combined with their unstable self-image and affective instability due to a marked reactivity of mood were identified as risk factors, which are also consistent with the diagnostic characteristics of BPD. Previous studies on adolescents or early adults also showed that these characteristics of BPD are strongly related to self-harm behavior [74]. This study did not consider the aspects of personality disorder, but further studies are needed to explore these aspects. The results of this study suggest that it is important to evaluate the negative self-image, irritability, and physical/sleep problems within the group experiencing depression and anxiety symptoms in order to understand the risk of self-harm. In addition to the findings of previous studies that negative self-image and irritability have associations with self-injury among adolescents [75,76], we found that physical and sleep problems are important risk factors in self-injury among adolescents. In particular, in the case of adolescents’ sleep problems, the relationship with suicidal ideation has been explored in Korea [77], but it is noteworthy that this study revealed the relationship between sleep problems and actual self-harm behavior in early adolescents.

Previous research states that as attempts to avoid negative affect become chronic, individuals experience that the aversive affect is more strong and frequent, making it harder to learn new and adaptive ways to process unpleasant emotions. Consequently, it creates a vicious cycle, where one repeatedly uses maladaptive methods involving avoiding negative emotional experiences [19]. Adolescence is a period known for vulnerability in affect regulation, and thus, it is a crucial time to develop coping mechanisms to regulate various emotions effectively [78]. Therefore, guiding teenagers to develop a healthy and available coping skill, as an alternative to self-injuring, will be helpful in effectively preventing and reducing adolescent self-injury. For intervention, applying treatments such as dialectical behavior therapy (DBT), which provides techniques to develop a positive self-image, interpersonal effectiveness, and emotion regulation skills [79], and cognitive behavioral therapy (CBT), which helps to modify negative self-image [80], can reduce self-injury among adolescents in the early stages and further decrease the risk of suicide. Moreover, Whitlock et al. [69] stated that adolescents form mutual relationships and experience social support on social media by sharing information on self-injury and seeking help. Such results suggest that it is essential to include elements enhancing social connectedness among others on top of reinforcing coping skills, in terms of intervention. Peer relationship becomes a key value in early adolescence, which is when the urge to belong to or find acceptance by peer groups becomes intense. Therefore, it is necessary to help adolescents deal with feelings of frustration and isolation by letting them share their feelings, and receive support from others through some means other than self-injury. To do so, it is necessary to prepare the school and home environments where they can express their emotional difficulties and find support, in advance.

This current study finds that low self-esteem did not significantly predict the incidence of self-injury. Prior studies have reported that self-esteem has some moderating effects on the occurrence and maintenance of self-injurious behavior [81]. Considering such findings, to confirm the effects of self-esteem on self-injury, it is essential to replicate this research by examining the relationships among intra- and interpersonal variables from multiple aspects in future studies.

This study has some limitations and suggestions for future studies. First, the surveys in this study targeted sixth graders in elementary school to third graders in middle school; however, the sample contained only a few elementary schoolers. Therefore, the results of this study reflect only a small amount of data of self-injury in younger adolescents. Secondly, the low value of internal consistency of the FASM should be noted as one of limitations in the present study. This might be due to the use of the subscale of the FASM; we only included the first part of the FASM consisting of 11 items, which might affect the low reliability of the measurement. Indeed, Cronbach’s alpha is known to be related to the number of items included [82,83,84]. Future studies should use the full range of the scale to consider the full characteristics of self-injurious behavior. According to Lloyd et al. [53], the reliability of all 39 questions of the FASM was reported as 0.88. Third, this study focused on some specific psychological factors that would have predicted self-injurious behavior in early adolescents; however, other previous studies suggested that various intra- and interpersonal variables show meaningful relationships with self-injurious behavior [4,38]. In addition, according to Laukkanen et al. [85], drug abuse such as frequent consumption of alcohol and legal drugs increased the risk of self-harm in adolescents aged 13–18 years. Although we investigated the use of alcohol during self-harm, the relationship between drug-related problems and self-harm was not addressed in this study. Future research should include more variables known to significantly affect self-injurious behavior and explore the predictive powers of such variables in detail. In addition, because prior studies find gender differences in risk factors of self-injurious behavior [86], future research may examine the risk factors predicting self-injury for each gender, thereby, providing some useful data for the prevention and treatment of self-injurious behavior. Finally, in this study, intrapersonal variables related to self-injurious behavior were measured using self-report questionnaires. However, according to a previous study using FMRI, adolescents with non-suicidal self-injury tended to show significantly stronger brain response in the amygdala and decreased activity in the frontal cortex to emotional stimuli [87]. Moreover, it has been reported that various forms of impulsiveness (e.g., affective, cognitive, and behavioral impulsivity) are related to adolescents’ self-harm behavior [88]. Therefore, in future research, it is necessary to additionally explore cognitive and temperamental characteristics (e.g., frontal lobe function and impulsiveness) related to adolescents’ self-harm by applying various research methodologies such as brain imaging research and experiments.

Despite the limitations of the present study, it has several implications. Given the high prevalence of self-harm behaviors in early adolescents, it provides a meaningful message for mental health professionals to acknowledge the importance of early detection and intervention for the prevention of adolescents’ self-harm. Acknowledging no proper assessment tool to detect, especially, early adolescents’ self-harm behaviors, it might be helpful to develop a sensitive screening tool reflecting high frequencies of self-harm behaviors reported by early adolescents. Based on our findings, clinical intervention strategies should focus on the modification of negative self-image, emotional dysregulation, and sleep problems using DBT and CBT.

## 5. Conclusions

This study shows that the prevalence of self-injurious behavior among adolescents in Korea was reported to be somewhat higher than in previous studies, and it can be regarded to reflect that self-injury has been recognized as a more common phenomenon in adolescents nowadays. As a result of the study, negative self-image, oversensitivity, and physical/sleep problems were identified as risk factors for adolescent self-harm. This implies the need to assess the risk of self-harm early and intervene through an evaluation focused on self-image and emotional instability in adolescents.

## Figures and Tables

**Table 1 ijerph-18-01955-t001:** Demographic characteristics.

Variables	SI Group (*n* = 166)	Non-SI Group (*n* = 350)	Total (*n* = 516)	*t* or *x*^2^	*p*
Age (*M/SD*) *	13.79 (0.87)	13.71 (0.90)	13.74 (0.89)	−0.95	0.34
Gender (*n*/%)				5.89	0.01
Male	58 (34.9)	154 (44.0)	212 (41.1)		
Female	99 (59.6)	162 (46.3)	261 (50.6)		
Grade (*n*/%)				0.19	0.97
6th	3 (1.8)	8 (2.3)	11 (2.1)		
7th	73 (44.0)	156 (44.6)	229 (44.4)		
8th	41 (24.7)	87 (24.9)	128 (24.8)		
9th	49 (29.5)	99 (28.3)	148 (28.7)		
Regions (*n*/%)				3.91	0.14
Metropolitan Areas	87 (52.4)	175 (50.0)	262 (50.8)		
Medium sized and small city areas	72 (43.4)	143 (40.9)	215 (41.7)		
Rural Areas	7 (4.2)	32 (9.1)	39 (7.6)		

*M* = mean, *SD* = standard deviation.

**Table 2 ijerph-18-01955-t002:** Methods and frequency of self-injurious behaviors during lifetime.

Items	Number of Students (%)	*M (SD)*	Range
Cutting or carving	32 (19.3%)	0.69 (1.70)	0~6
Hitting	56 (33.7%)	1.01 (1.82)	0~6
Pulling hair	47 (28.3%)	0.66 (1.30)	0~6
Self-tattooing	0 (0.0%)	0 (0.00)	0~0
Picking at a wound	17 (10.2%)	0.25 (0.97)	0~6
Burning	1 (0.6%)	0.00 (0.07)	0~1
Inserting objects under nails or skin	7 (4.2%)	0.14 (0.84)	0~6
Biting	116 (69.9%)	2.24 (2.17)	0~6
Picking areas to draw blood	8 (4.8%)	0.13 (0.73)	0~6
Scraping skin to draw blood	33 (19.9%)	0.62 (1.48)	0~6
Erasing skin	23 (13.9%)	0.44 (1.47)	0~12

**Table 3 ijerph-18-01955-t003:** Characteristics of self-injurious behaviors.

Characteristics of SIB	Number of Students (%)
Duration of contemplation before SIB	
Had not contemplated SIB before	121 (72.9%)
For a few minutes	27 (16.3%)
For less than an hour	6 (3.6%)
For more than an hour and less than 24 h	2 (1.2%)
For more than a day and less than a week	3 (1.8%)
For over a week	7 (4.2%)
Alcohol or drug consumption	
Had used alcohol or drugs at the time of SIB	2 (1.2%)
Physical pain during SIB	
No pain	106 (63.9%)
Mild pain	44 (26.5%)
Moderate pain	14 (8.4%)
Severe Pain	2 (1.2%)
Intent to die	
Had had suicidal intention	18 (10.8%)

SIB = self-injurious behavior.

**Table 4 ijerph-18-01955-t004:** Correlations of variables related to self-injurious behaviors.

Variables related to SIB		1	2	3	4	5	6	7	8
1	SIB frequency	1							
2	Number of medical treatments	0.07	1						
3	Number of SIB methods	0.84 **	0.04	1					
4	Moderate SIB frequency	0.63 **	0.18 *	0.50 **	1				
5	Mild SIB frequency	0.94 **	0.01	0.80 **	0.34 **	1			
6	Duration of contemplation before SIB	0.01	0.01	0.00	0.06	−0.00	1		
7	Physical pain during SIB	0.46 **	−0.03	0.39 **	0.28 **	0.44 **	0.44 **	1	
8	Age of onset	0.01	−0.02	0.14	0.11	−0.02	0.35 **	0.18	1

SIB = self-injurious behavior; ** *p* < 0.01; * *p* < 0.05

**Table 5 ijerph-18-01955-t005:** Correlations of related variables.

Measures		1	2	3	4	5	6	7	8	9	10	11
1	Depression (CDI)	1	0.93 **	0.82 **	0.75 **	0.73 **	0.47 **	0.51 **	0.55 **	0.73 **	–0.81 **	0.54 **
2	Negative Self-image	0.89 **	1	0.69 **	0.57 **	0.69 **	0.47 **	0.50 **	0.47 **	0.69 **	–0.76 **	0.50 **
3	Interpersonal problems	0.79 **	0.53 **	1	0.43 **	0.55 **	0.33 **	0.41 **	0.34 **	0.58 **	–0.71 **	0.41 **
4	Negative affect/somatic concerns	0.71 **	0.46 **	0.44 **	1	0.58 **	0.37 **	0.36 **	0.57 **	0.53 **	–0.55 **	0.42 **
5	Anxiety (RCMAS)	0.55 **	0.51 **	0.36 **	0.43 **	1	0.80 **	0.76 **	0.66 **	0.89 **	–0.62 **	0.45 **
6	Worry	0.45 **	0.41 **	0.30 **	0.35 **	0.87 **	1	0.52 **	0.34 **	0.61 **	–0.45 **	0.27 **
7	Oversensitivity	0.39 **	0.36 **	0.25 **	0.29 **	0.77 **	0.56 **	1	0.35 **	0.58 **	–0.39 **	0.34 **
8	Physical sleep problems	0.38 **	0.30 **	0.25 **	0.38 **	0.67 **	0.49 **	0.35 **	1	0.48 **	–0.44 **	0.44 **
9	Negative affect/Attention problem	0.53 **	0.52 **	0.34 **	0.37 **	0.86 **	0.63 **	0.59 **	0.46 **	1	–0.62 **	0.38 **
10	Self-Esteem (RSES)	−0.67 **	−0.66 **	-0.53 **	−0.37 **	−0.53 **	−0.41 **	−0.40 **	−0.34 **	−0.54 **	1	−0.35 **
11	SIB frequency	–	–	–	–	–	–	–	–	–	–	1

CDI = Child Depression Inventory; RCMAS = Revised Children’s Manifest Anxiety Scale; RSES = Rosenberg Self-Esteem Scale; SIB = self-injurious behavior; Upper triangle contains correlations in SIB group (*n* = 166), lower triangle contains correlations in non-SIB (*n* = 350) group; ** *p* < 0.01.

**Table 6 ijerph-18-01955-t006:** Group differences in psychological characteristics.

Variables	SI Group (*n* = 166)	Non-SI Group (*n* = 350)	Total (*n* = 516)	*t*	*p*
CDI Total	15.63 (7.88)	10.33 (5.84)	12.04 (7.01)	7.71	0.00
Negative self-image	7.09 (4.14)	4.46 (3.15)	5.31 (3.70)	7.25	0.00
Interpersonal problems	4.13 (2.42)	3.14 (2.11)	3.46 (2.26)	4.52	0.00
Negative affect/Somatic concerns	3.87 (2.18)	2.55 (1.64)	2.97 (1.93)	6.89	0.00
RCMAS Total	12.39 (6.43)	6.85 (5.41)	8.63 (6.31)	9.60	0.00
Worry	3.77 (2.03)	2.29 (2.03)	2.76 (1.93)	7.72	0.00
Oversensitivity	2.66 (1.68)	1.48 (1.47)	1.86 (1.63)	7.75	0.00
Physical sleep problems	2.12 (1.61)	1.16 (1.24)	1.47 (1.44)	6.79	0.00
Negative affect/attention problem	3.83 (2.73)	1.91 (1.92)	2.53 (2.38)	8.12	0.00
RSES Total	27.95 (5.91)	31.38 (4.96)	30.28 (5.52)	–6.88	0.00

CDI = Child Depression Inventory; RCMAS = Revised Children’s Manifest Anxiety Scale; RSES = Rosenberg Self-Esteem Scale.

**Table 7 ijerph-18-01955-t007:** Binomial logistic regression analysis for variables predicting self-injurious behaviors.

Variables	B	SE	Wald	df	*p*	Exp (B)
Step 1						
Negative affect/attention problem	0.34	0.04	56.25	1	0.00	1.40
Constant	–1.64	0.16	96.63	1	0.00	0.19
Step 2						
Negative self-image	0.11	0.03	9.37	1	0.00	1.12
Negative affect/attention problem	0.23	0.05	17.14	1	0.00	1.26
Constant	–1.99	0.21	90.22	1	0.00	0.13
Step 3						
Negative self-image	0.10	0.03	7.48	1	0.00	1.11
Physical sleep problems	0.22	0.08	6.45	1	0.01	1.24
Negative affect/attention problem	0.18	0.06	9.53	1	0.00	1.20
Constant	–2.13	0.22	92.22	1	0.00	0.11
Step 4						
Negative self-image	0.09	0.03	6.07	1	0.01	1.10
Oversensitivity	0.18	0.08	5.22	1	0.02	1.20
Physical sleep problems	0.20	0.08	5.28	1	0.02	1.22
Negative affect/attention problem	0.11	0.06	3.09	1	0.07	1.12
Constant	–2.26	0.23	94.43	1	0.00	0.10

Stepwise method: forward entry.

## Data Availability

Data available on request due to privacy/ethical restrictions.

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
