# Peer review of "Self-Injurious Behavior in Community Youth"

_ijerph, 2021, doi:10.3390/ijerph18041955_

Round 1
Reviewer 1 Report
Thank you kindly for the opportunity to review this manuscript. The topic investigated is definitely one worthy of attention, and this study adds to the field by surveying a large sample of Korean school students. Findings, particularly around the relatively high percentage of participating reporting lifetime history of engaging in self-injury and associated factors, are interesting and worth sharing with international audiences. However, I am afraid that the manuscript suffers from too many flaws to be able to recommend it for publication in this journal.
Nevertheless, I provide some comments and recommendations for improvement should the authors consider revising the paper and submitting it elsewhere:
- As an overall comment, I felt the language used in the article did not meet the required standards. This is most apparent in the Abstract but also throughout the manuscript. I would strongly encourage authors to submit the paper for professional editing, also with the view of creating a somewhat more concise Introduction and Discussion. I also encourage authors to amend some terms such as “having committed self-injurious behavior” and “self-injurer” as they may be perceived as stigmatizing or labelling.
- In methods, the authors state “The participants included sixth graders in elementary school and first to third graders in middle school of both genders”, but demographic info in Table 1 shows frequencies for 6th , 7th , 8th, and 9th This could be better explained by making it clear that grades 7-9 are part of middle school ion Korea.
- The reported Cronbach alpha for Functional Assessment of Self-Mutilation is very low – this raises concerns about the reliability of the instrument and should be acknowledged as a limitation.
- In Results, the claim that “significant gender difference was apparent; more female students had self-injured than had male students” is not easily seen by looking at the data in Table 1 which at the moment shows that among SI group, there were proportionally more females than there were in non-SI group. This needs to be reworded.
- A particularly noteworthy result is a very high frequency of participants who had used “biting” as a SIB method – this would need to be explored more in Discussion and placed in the context of existing literature on the topic.
- In Table 2, it is unclear what the data in column “Frequency” represents – is that the combined number of self-injurious acts performed by the students who endorsed this answer? If so, this might be redundant information, and M, SD and perhaps min and max are more pertinent to the understanding of the extent of these behaviours.
- In addition, other results would be worth adding to Table 2 (currently only presented in a narrative format) – prior contemplation, severity, degree of pain, use of alcohol or drugs, the suicidal nature of the act.
- Table 5: It is unclear why a hierarchical logistic model was used, there is no rational for this provided nor are results interpreted in this light. I would suggest replacing it with a simpler ‘entry’ method, otherwise provide an explanation / interpretation of this.
- The title of the section “Risk factors predicting self-injurious behaviours” is not accurate as the study design and analyses do not allow for making inferences about causality.
Reviewer 2 Report
I have the following comments for the authors to address and I am happy to review this paper again.
1) The authors stated "Meta-analysis studies have reported 4.1–41.5% of life prevalence of self-injurious behavior in adolescence, varying with samples and methods of evaluation. They estimate overall life prevalence at 25% [3]." Reference 3 is not based on adolescents. It is more relevant to refer to a meta-analysis focusing on adolescents. The authors should go to Pubmed to search "Global Lifetime prevalence and Deliberate Self-Harm and Non-Suicidal Self-Injury in Children and Adolescents and meta-analysis" and look for more recent and relevant prevalence.
2) The authors stated "Many studies have identified depression as the 122
risk factor predicting self-injurious behavior in Korea and abroad". One of the key factor that is being ignored or neglected is borderline personality disorder in development. The authors should go to Pubmed and search for studies on "risk factors for repeated suicide attempts in multi -ethnic asian" and check whether borderline personality disorder is a risk factor. Then the authors should explain what is borderline personality disorder. The authors should go to Pubmed and look for studies that discuss the core symptoms of borderline personality disorder by searching: "Borderline personality disorder and affect dysregulation, self-disturbances, and behavioral and interpersonal dysregulation".
3) The authors should add the following limitation: This study did not like into the relationship between substance misuse and self harm behaviour.
4) The authors stated "Future research should include more variables known to significantly affect self-injurious behavior and explore the predictive powers of such variables in detail." I agree but need to be more specific. The authors should go to Pubmed and look for studies on "Functional near infrared spectroscopy (fNIRS) and depression and borderlne personality disorder". Please comment on the frontal lobe impairment that is associated with impulsivity.
5) Please discuss how psychotherapy e.g. Dialectical behavior therapy can help to reduce self harm.
Reviewer 3 Report
Thank you for providing me the opportunity to review this manuscript, which reports the findings of a study on self-injury in community youth in Korea. The topic of the study is certainly in need of high-quality research. However, there are conceptual unclarities, and especially the discussion section can be strengthened. I detail my concerns below.
The introduction starts by defining self-injurious behaviour. However, it would be good to state whether or not this includes self-harming behaviour with suicidal intention. The list of references includes many papers that are concerned with non-suicidal self-injury, which seems to suggest that this is the focus of the study. However, looking at the statement on page 2, and the data collection, it seems that the study is concerned with self-injury with or without suicidal intention. It would be good if authors could be clear about this at the onset of the paper.
In addition, there are many studies about self-harm irrespective of suicidal intention conducted in the UK and internationally, for example by the research unit in Oxford. None of these studies have been cited in this paper, which may contribute to a perception that the study is focused on non-suicidal self-injury.
Throughout the manuscript, please avoid the expression “to commit” suicide as this may refer to a criminal act. In international literature it is more common to use wordings such as “to die” by suicide, or similar.
Line 38: “…correlated with various psychiatric symptoms”. Can you provide an example?
Line 40: “…it can increase the risk of suicide”. Please specify how much it can increase the risk of suicide. Please provide a reference for this statement.
Lines 41-51: The percentages about the prevalence of self-injurious behaviour have these been reported by national representative samples or selective samples, such as from one school, or self-selected participants? (in other words, how representative are these percentages?)
Lines 57-59: The text says that self-injury has become a serious problem. Does this imply that there has been an increase over the years? Is there evidence for this? Or has awareness in society increased?
Line 80-81: Is not clear if you refer to the same adolescents in the past, or to adolescents in general. Please rephrase for clarity.
Lines 147-152: I am not sure if I understand the aims of the study correctly. What do authors mean with “substantial fundamental references”?
The study aims to increase our understanding of the risk and predictive factors of self-injury; however, the aim does not include preparation of intervention or prevention strategies?
Line 154: A total of 582 participants had provided consent; but how many potential participants (students) had received the study information?
Line 181: “the amount of pain they felt”: before, during or after the self-injury? Physical pain? Psychological pain?
Line 185: A Cronbach’s alpha of .59 is rather low, especially if you want to use the instrument in a model of analysis. I would see this as a limitation of the study.
Discussion: In the introduction/aims authors stated that this study was needed to generate new knowledge and insights into risk and predictive factors. However, reading the discussion, I have the impression that most findings had already been reported by previous research. If this impression is not correct, then authors might emphasize a bit more what the new findings are.
In the introduction authors had also introduced several explanatory models of self-injurious behaviour. Have these models been useful for the study? Do the findings match with the models?
Lines 301-303: please see earlier question about representativeness of the sample.
Line 425: Implications are vague. Can you be more specific about potential clinical or public health implications.
Line 430: First sentence of the conclusions does not seem to be related to this study.
Overall, I can see merit in this study, especially for service providers and clinicians in this field. Hence, I hope that these few comments may help the authors revising the manuscript. Good luck.
Round 2
Reviewer 1 Report
I would like to thank the authors for undertaking a significant revision of the paper, including professional editing. It has definitely improved its quality. There remain, however, a few outstanding issues that I kindly ask them to consider before I can recommend the paper for publication:
INTRODUCTION
- The third sentence lists various types of NSSI, however, the list is not exhaustive, so please expand it.
- Replace the term “life prevalence” with “lifetime prevalence”.
- In the introduction, you reference a study by Korean Ministry of Education (13) that found prevalence of SIB among middle school and high school students to be around 6-7%. However, in the Discussion, you reference Korean studies that report of prevalence around 20%., and your results were at over 30%. The study described in the Introduction is from 2018, so the vast differences cannot be attributed to the recent increases in SIB – please provide some explanation around this in the paper.
- The first sentence added to the Abstract in red text duplicates the same message as already stated above.
- In the sentence “adolescents in local communities reported milder forms of self-injury”, it is unclear in comparisons to whom?
- Please rephrase the last sentence finishing with “to hinder the risk of self-harm” as this verb is not commonly used in relation to risk factors.
RESULTS
- Table 2 – Thank you making the recommended changes from ‘frequency’ to reporting M (SD) and range values. However, I am afraid the table still does not read very easily. I suggest changing the title to add the timeframe within which these behaviours were measured – e.g. “Methods and frequency of self-injurious behaviours during lifetime” or similar.
- Table 3 – thank you for including Table 3 into results, it is definitely easier for the reader to grasp these results when presented in a tabular format. However, you still need to provide a bit of commentary around the main findings in the text.
- I find the result that 73% of the sample did not contemplate SIB at all, not even for a few minutes, before engaging in it, very surprising. Also, the fact that only 1.2% had consumed alcohol or drugs while engaging in SIB. There is currently no discussion around these results, but it is very important to add some.
- Finally, please compare your finding that 10.85% had suicidal intention while SIB into context of similar literature; the overlap between suicide attempts and SIB are a well-explored topic.
- In Table 5, please replace the acronyms for scales with the two main factors – Depression and Anxiety-, and then list the sub-scales more clearly so it is evident they are indeed sub-factors of the larger ones.
DISCUSSION
- Seeing that this manuscript was submitted to an international Journal, it would be important to consider the role of cultural factors on the observed findings – e.g. does literature note any significant differences in the prevalences, types and correlates of SIB in the Western vs Asian countries?
- Please place the text describing Limitations in a separate paragraph.
GENERAL COMMENTS
- Please use consistently numbers with either 1 or 2 decimals.
- Please reposition tables throughout the manuscript so they are placed immediately after first mentioned in text. For example, on page 8, you first discuss Table 5 followed by text about Table 6, and only then you show Table 5, while Table 6 is placed in the middle of text pertaining to risk factors related to SIB. This interrupts the flow.
